# Compliance to a Gluten-Free Diet in Swedish Children with Type 1 Diabetes and Celiac Disease

**DOI:** 10.3390/nu13124444

**Published:** 2021-12-13

**Authors:** Hanna Söderström, Julia Rehn, Matti Cervin, Cathrine Ahlstermark, Mara Cerqueiro Bybrant, Annelie Carlsson

**Affiliations:** 1Department of Clinical Sciences Lund, Lund University, 221 00 Lund, Sweden; ju0854re-s@student.lu.se (J.R.); matti.cervin@gmail.com (M.C.); Annelie.carlsson@med.lu.se (A.C.); 2Skåne University Hospital, 221 85 Lund, Sweden; cathrine.ahlstermark@skane.se; 3Department of Women’s and Children’s Health, Division of Pediatric Endocrinology, Karolinska Institutet, 171 76 Stockholm, Sweden; mara.bybrant@ki.se

**Keywords:** type 1 diabetes, celiac disease, children, gluten free diet, metabolic control, HbA1c, diabetic ketoacidosis

## Abstract

Children with type 1 diabetes (T1D) are at increased risk of celiac disease (CD). The replacement of insulin in T1D, and the exclusion of gluten in CD, are lifelong, burdensome treatments. Compliance to a gluten-free diet (GFD) in children with CD is reported to be high, while compliance in children with both diseases has scarcely been studied. To examine compliance to a GFD in children with both T1D and CD, we analyzed tissue transglutaminase IgA-antibodies (tTGA). Moreover, associations between compliance and age, sex, glycemic control, ketoacidosis (DKA), body mass index (BMI), and time of CD diagnosis were investigated. Of the 743 children diagnosed with T1D in southern Sweden between 2005 and 2012, 9% were also diagnosed with CD. Of these, 68% showed good compliance to a GFD, 18% showed intermediate compliance, and 14% were classified as non-compliant. Higher age, poorer HbA1c, and more DKAs were significantly (*p* < 0.05) associated with poorer compliance. In conclusion, we found that compliance to a GFD in children with T1D and CD is likely be lower than in children with CD only. Our results indicate that children with both T1D and CD could need intensified dietary support and that older children and children with poor metabolic control are especially vulnerable subgroups.

## 1. Introduction

Children with type 1 diabetes (T1D) have an increased risk of being diagnosed with celiac disease (CD) compared to the general population [1]. The estimated prevalence of CD in children is approximately 1% [2], while the prevalence of CD in children with T1D ranges between 2.4 and 16.3% [3], reflecting national and international differences in CD diagnostics and CD risk [1]. The prevalence of CD in Swedish children with T1D is around 10% [4,5]. T1D and CD are chronic diseases that share genetic risk factors such as HLA DQ2 and DQ8 [6]. The incidence of both diseases has increased during the last few decades [7,8,9], pointing towards possible environmental risk factors [10].

The necessary replacement of insulin in T1D and the total exclusion of dietary gluten in CD are lifelong, burdensome treatments, and the double burden for children with both diseases is reported to be difficult [11,12]. A diet with a low glycemic index is recommended for patients with T1D. However, GFD foods often have a high glycemic index [13,14] and, accordingly, children with CD have a higher dietary glycemic index in their diet [15,16]. For children with both T1D and CD, it may be difficult to find GFD foods that are low in sugar and high in fiber to keep a good metabolic control [11,17]. Even without T1D, a GFD can be hard to strictly follow. In one study, accidental gluten intake was observed in 86% of children with CD that aimed to keep a strict GFD [18].

In patients with untreated CD, the intestinal mucosa is damaged by an immunogenic reaction to gliadin. This reaction is measured by autoantibodies (tissue transglutaminase IgA antibodies (tTGA), endomysial antibodies (EMA), or deamidated gluten peptides (DGP)). A strict GFD usually resolves enteropathy and levels of tTGA are shown to be closely related to the degree of intestinal mucosa damage [19,20]. Thus, significant increases in tTGA levels can be used to detect a rise in CD activity [21]. The time needed to detect a healed mucosa through tTGA levels can vary, but for most GFD compliant CD patients, tTGA normalizes in around a year [22]. Interestingly, children with both CD and T1D on a GFD are reported to have a longer tTGA normalization period, ranging between 1 and 2 years or even longer [22,23]. 

In some children, CD is discovered because of typical symptoms like abdominal pain, failure to thrive, or diarrhea, but a large proportion seem to lack typical symptoms [24]. A Swedish screening study for CD in 12-year-olds from the general population showed that two-thirds of children with CD had an undiagnosed disease [25]. Likewise, in Swedish children with T1D, CD is often asymptomatic [5]. This is shown in another Swedish study on children with T1D that found 0.7% to be diagnosed with CD prior to T1D diagnosis, while 3% had a silent, screening-detected CD at T1D diagnosis, and annual screenings then led to a cumulative CD incidence that reached 10% after five years [4]. 

Around 90% of Swedish children with CD, including screening-detected cases, comply to a GFD [18,26], but there are few studies on compliance to a GFD in children with additional T1D. Most studies are small (including 9 to 35 children) and with somewhat conflicting results. The majority of these studies reported compliance to be relatively high, 63–100% [17,23,27,28,29], but two studies reported compliance to be only 30–36% [30,31]. In most studies, compliance was associated with better glycemic control [28,29] and/or growth [29,31]; only one found no association between compliance and growth [30]. We found one larger registry study on 608 double-diagnosed children which also reported good compliance to be associated with both better glycemic control and improved growth, but this study only found a good dietary adherence in 36% of the children [32]. Moreover a randomized controlled study comparing 15 double-diagnosed children on a GFD to 15 double-diagnosed children on a normal diet found children on a GFD to have better glycemic control after 12 months [33].

Because of the limited number of studies and small sample sizes, more knowledge on compliance to a GFD in children with T1D and CD is needed. Inconsistent results on whether or not compliance is associated with glycemic control and growth also show the need for more research. In this study, we examined compliance to a GFD in all children with T1D and CD from southern Sweden. All participants were screened for CD at T1D diagnosis and then screened annually for up to 10 years. Associations between compliance and a range of sociodemographic and clinical factors were examined. 

## 2. Materials and Methods

### 2.1. Participants and Setting

This study was approved by the ethical committee in Lund in 2014 (Dnr:2014/476) with a complementary approval in 2020 (Dnr: 2020-04152). We collected data from medical records for all children, aged 0–18, in Skåne county, Sweden, diagnosed with T1D between 2005–2012. The diagnostic criteria used for T1D patients in Sweden subscribe to the diabetes classification of the American Diabetes Association [34], whereas all included children with a CD diagnosis were diagnosed by confirmed small intestinal biopsy. In general, all children diagnosed with T1D were screened for CD at T1D diagnosis and in most cases annually thereafter. A CD diagnosis either pre- or post-T1D diagnosis was an inclusion criterion for the study. 

Between 2005 and 2012, 743 children in Skåne county were diagnosed with T1D. Of these, 64 (9%) had CD proven by a biopsy. The follow-up time ranged from 1 to 10 years. Information on date of birth, sex, and age at both T1D and CD diagnosis were collected from the medical records. Longitudinal data for HbA1c and BMI were collected from the national Swedish quality register, SWEDIABKIDS, complemented by data from medical journals at the included hospitals in Skåne. Information about the CD diagnoses was collected from the diagnostic registry by Statistics Sweden and confirmed by the medical journals of the children. Levels of antibodies (tTGA and EMA) were collected from medical journals at the included hospitals. 

### 2.2. Compliance

#### 2.2.1. tTGA

We used tTGA, a recommended serological measure for analyzing adherence to a GFD, for assessing compliance to a GFD [21]. tTGA was measured through blood samples and analyzed in a EliA Celikey IgA system from Thermo Fisher Scientific (Legal Manufacturer: Phadia AB, Uppsala, Sweden). An outcome equal to 10 U/mL or above was classified as positive [35]. However, EMA was still in use at the very beginning of the study and a few measurements were reported in EMA instead of tTGA. In accordance with a study that screened for CD in the general population using and evaluating both EMA and tTGA [36], as well as regional laboratory guidelines [37], a cut-off of <10 (1/10) was used for EMA as well.

#### 2.2.2. Classifying Compliance 

To evaluate adherence to a GFD, all patients with ≥3 serological results more than two years after CD diagnosis were divided into four compliance categories. 

Insufficient data (<2 serological results more than two years after CD diagnosis).Good compliance (all tTGA values < 10).Intermediate/varying compliance (≥1 tTGA value being positive and ≥1 tTGA value being negative).Non-compliance (all reported tTGA values ≥ 10).

A two-year period after the CD diagnosis was used because of prior reports that the antibody normalization period for children with T1D is longer [22,23].

All patients with ≥5 measurements that had just one deviating value (either positive or negative) were classified in accordance with the majority of values. 

### 2.3. Metabolic Control and Growth

#### 2.3.1. HbA1c and DKA

To evaluate metabolic control, we used glycemic control and ketoacidosis (DKA). To evaluate glycemic control, we used glycated hemoglobin (HbA1c) in blood, a measure of the blood glucose levels of the 2–3 preceding months. HbA1c was analyzed using the Capillarys 3 TERA Hemoglobin A1c Kit-program, accredited and used at Skanes uUniversity hospital, analysed at Labmedicin, “Laboratorium”, 105 02 Malmö. where separation is on the capillaries. Separation is optimized to eliminate interferences from variants of hemoglobin, pre-HbA1c and carbamylated hemoglobin. The instrument analyzes 68 samples per hour. Detection is at 415 nm. 

DKA was defined as pH below 7.30 and we compared those that reexperienced DKAs with those that did not. 

#### 2.3.2. Growth

To evaluate growth, we used the standard deviation score of body mass index (SDS-BMI) [38]. 

### 2.4. Statistical Analysis

All statistical analyses were conducted using IBM SPSS Statistics version 25. To analyze the degree of compliance, we estimated the proportion of participants being classified into each compliance group (good, intermediate, non-compliant). To examine associations between compliance and metabolic control, growth, age, sex, DKA, and whether CD was diagnosed prior to or after T1D, we conducted a line of univariable ordinal regression models with compliance (good, intermediate, non-compliance) being the dependent variable and each of the above variables being entered as independent variables. Ordinal regression allowed us to respect the ordinal nature of the three compliance groups. An alpha level of 0.05 was used as an indicator of statistical significance. All variables that showed a statistically significant association with compliance were entered into a final ordinal regression model. In this model, we made an a priori decision to include age and sex as independent variables, as these were deemed to be possible confounders. Compliance was entered as the dependent variable in the analyses as we could not derive clear causal patterns from the data.

## 3. Results

### 3.1. Participant Characteristics and Degree of Compliance

Of the 64 children that met the inclusion criteria, information on study variables could be retrieved for 60 children (94%). See Figure 1 for further information regarding patient inclusion. Medical records about HbA1c, BMI, and DKA were not available for the four patients that were excluded. Thus, a sample of 60 patients with T1D and CD was identified. Of the children, 32 (53%) were female and 28 (47%) were male. The mean age at the T1D diagnosis was 8.87 years (SD = 4.50; range = 1–17 years). Of these, 19 (32%) were diagnosed with CD prior to T1D and the rest had it detected at or after the T1D diagnosis. See Table 1 for full information on the demographic and clinical variables in the full sample and in the four compliance subsamples.

Following the criteria used to classify degree of compliance, 10 patients (17%) had insufficient data due to too few follow-ups or because not enough time had elapsed after the CD diagnosis, resulting in 50 patients being included in the final analyses regarding compliance. The majority, 34 patients (68%), were classified as having good compliance, 9 patients (18%) as having intermediate compliance, and 7 patients (14%) as being non-compliant. Forty-four patients (73%) had tTGA values on at least five follow-up occasions. Thirty patients normalized their antibody levels: 23 (77%) within 1 year and another 6 (23%) within the second year. 

### 3.2. Associations between Clinical/Demographic Variables and Compliance

We found no difference in compliance between those that received a CD diagnosis prior to T1D compared to those who received their CD diagnosis at or after T1D diagnosis (*p* = 0.91). 

#### 3.2.1. Compliance and HbA1c

The mean HbA1c value during the follow-up study period had a significant association with compliance (OR = 1.09, CI_95%_ = 1.02–1.16, *p* = 0.008) such that higher mean HbA1c values increased the odds of being classified as having poorer compliance, see Figure 2. 

#### 3.2.2. Compliance and Age

In a univariable model, age was significantly associated with compliance (OR = 1.41, CI_95%_ = 1.14–1.75, *p* = 0.002) such that older age was associated with increased odds of being classified as having poorer compliance, see Figure 2. In a univariable model, sex was not statistically significantly associated with compliance (OR = 3.57, CI_95%_ = 0.96–13.22, *p* = 0.057), but it trended towards significance with girls showing increased odds of being classified as having poorer compliance.

#### 3.2.3. Compliance and BMI

Median SDS-BMI for the whole group was below −0.5 at baseline and changed to +0.4 at year one and remained positive (0.5–0.7) throughout the ten-year follow-up period (see Table 1). Mean SDS-BMI was not significantly associated with compliance (*p* = 0.36) but there was a trend towards a higher SDS-BMI with higher compliance.

#### 3.2.4. Compliance and DKA

To analyze whether there was an association between DKA and compliance, we divided participants into those who experienced at least one DKA during the follow-up period (*n* = 10) and those who did not (*n* = 37). For three participants with a compliance classification, no information about DKA was available. The univariable ordinal regression indicated a significant association between DKA and compliance (OR = 6.22, CI_95%_ = 1.53–25.33, *p* = 0.011, *n* = 47), indicating that those who had a DKA had increased odds of being classified as having poorer compliance, see Figure 1. DKA was also significantly associated with poorer compliance (OR = 7.39, CI_95%_ = 1.44–37.83, *p* = 0.016, *n* = 47) when accounting for age and sex.

#### 3.2.5. Full Model

In line with our statistical plan, we conducted a model that included age, sex, and HbA1c as independent variables and compliance as the dependent variable. Results showed that age and HbA1c were significantly associated with compliance, and the results fully aligned with those reported above: age (OR = 1.36, CI_95%_ = 1.07–1.73, *p* = 0.011); sex (OR = 4.08, CI_95%_ = 0.85–19.73, *p* = 0.080); and HbA1c (OR = 1.10, CI_95%_ = 1.02–1.17, *p* = 0.011). The pseudo *R*^2^ (Nagelkerke) of this model was 0.420, indicating that 42% of the variance in compliance could be explained by variance in the independent variables. Lastly, we added DKA to the above model. DKA was not entered in the previous model because of missing data for three participants. In this model (*n* = 47), only age (OR = 1.48, CI_95%_ = 1.13–1.93, *p* = 0.004) was statistically significantly associated with compliance, while HbA1c (*p* = 0.60), DKA (*p* = 0.06), and sex (*p* = 0.11) were not. The model explained 46% (Nagelkerke) of the variance in compliance.

To further examine the associations between DKA, HbA1c, compliance, and whether collinearity problems were present (i.e., that two independent variables were highly associated), we ran an independent samples t-test which showed that there was a clear and statistically significant association between having experienced a DKA and having poorer mean HbA1c (DKA, M = 72.71 (SD = 12.72); no DKA, M = 58.94 (SD = 7.90); t(45) = 4.26, *p* < 0.001). Thus, both HbA1c and DKA were significantly associated with compliance when accounting for age and sex, while neither HbA1c nor DKA were significantly associated with compliance when they were included in the same model, which indicates that HbA1c and DKA are associated and may suppress each other when they are entered in tandem in a statistical model.

## 4. Discussion

In this study, we found compliance to a GFD in children with both CD and T1D to be at 68%, which is considerably lower than the 85–96% that has been found in large samples of Swedish children with CD only [18,26] We also found poor compliance to be associated with being older, having poorer glycemic control, and a higher probability of experiencing DKA. Whether CD diagnosis was present before T1D or detected at or after T1D diagnosis was not associated with the level of compliance. 

Our compliance rate (68% good compliance) is similar but somewhat lower compared to estimates from other studies in children with T1D and CD, where compliance, assessed by antibodies, dieticians’ assessment, and/or questionnaires, was found to be around 80% [17,23,27,28,29]. Although smaller (between 9 and 35 children) and with fewer longitudinal follow-up assessments, these studies, conducted between 2002–2016, provide estimates in line with ours. Taken together, this indicates that a majority of children with T1D and CD show a fairly good compliance to a GFD. 

In contrast, two small studies with 9 and 20 children [30,31], conducted in 2008 and 1999, respectively, and one large register study with 608 children [32], conducted in 2019, found substantially lower compliance: between 30–44%. Reasons for these differences are unclear but could be explained by older data, different methods to establish compliance, and representativity of the analyzed samples. In 1995, GFD foods were less available and less variable, making it harder to follow a GFD. Two of the studies were old and also quite small. In the larger and newer study, data collection started as early as 1995. Also, a multicenter design where antibody testing was performed by different laboratories with different types of tTGA tests was used, which makes a direct comparison difficult [21]. Another limitation with the registry study is that the majority (62.7%) of the registered children with T1D were excluded. We also speculate that screening to detect CD in these T1D populations might have been scarce, since only 1.36% of the children with T1D in their material were found to have a double diagnosis, leaving a less representative sample. 

To understand whether the presence of T1D in addition to CD affects compliance, it is important to compare the compliance of children with a double diagnosis to the compliance of children with CD only. In Sweden, all patients with CD—symptomatic or asymptomatic—are asked to keep a strict GFD. A Swedish study from 2014 showed that 96.8% of children with CD adhered to a strict GFD [18] and another Swedish study from 2015 showed that normalized tTGA levels were present in 85% after one year [26], although the true compliance rate was probably higher since one year may be a short follow-up time for complete tTGA normalization [26]. Thus, although children with T1D and CD in our study and many others show fairly good compliance, compliance seems to be lower than in children with CD only. This is supported by a Canadian study from 2017 on 487 pediatric patients diagnosed with CD, which found compliance to a GFD to be significantly different in patients with T1D: 77% compared to 89% in patients without T1D. Additionally they found a significantly longer time to tTGA normalization in CD children with T1D: 1204 days compared to 403 days in patients without T1D [22]. In our study, 77% of the children that normalized their tTGA levels reached normalization within one year, while 23% needed two years. Thus, if normalized tTGA levels one year after the CD diagnosis had been required to be classified as having good compliance, compliance levels would have been markedly lower. That many children with T1D and CD seem to need more than one year to normalize tTGA is supported by previous research [22,39] and may be important to consider when examining compliance to a GFD in children with T1D. Whether this difference depends on the burden of management and control of two heavy treatments, or on the physiological ability to neutralize tTGA antibodies in children with T1D, remains to be answered. 

Our results with a lower compliance in children with T1D and CD compared to children with CD only is important as it indicates that children with both diseases could need intensified support. This support could, for example, be more frequent visits with a dietician who specializes in the specific challenges that comes with monitoring both T1D and CD simultaneously. These results may also suggest children with both T1D and CD as a group suitable for trying promising new treatments like transglutaminase 2 inhibitors [40].

In our study worse glycemic control and the risk of experiencing DKAs was associated with poorer compliance. That better compliance is associated with a better glycemic control is in line with previous studies [28,29,32], and the only, to our knowledge, randomized controlled study (RCT) on the topic to date also found that a GFD resulted in better glycemic control [33]. In contrast with our results, the only other study on DKAs we found described no association between compliance and DKA [32]. 

The associations between poorer compliance and poorer glycemic control and more DKAs could be related to a possibility of active CD negatively influencing metabolic control [41]. Another, and perhaps more plausible explanation, is that individual differences in overall compliance to healthcare treatments and advice may explain these associations. Children who tend not to adhere to a GFD perhaps are less likely to adhere to their T1D regimen, resulting in both poorer compliance to a GFD, poorer metabolic control, and more DKAs. 

We also found higher age to be associated with poorer compliance which has been reported previously [32]. Several studies have reported older children with T1D to have worse metabolic control than younger children [42,43]. This makes age a tempting explanation for the association of worse metabolic control to poor compliance. However, we found both age and metabolic control to be independently associated with compliance, indicating that age alone does not explain the link between compliance and glycemic control.

We did not find any significant association between compliance and BMI, although there was a trend towards higher BMI being associated with better compliance. The only other study on BMI and GFD compliance in T1D patients [31] reported the same trend. Other studies [23,27,44] have reported an increased growth velocity in children after being diagnosed with CD and introduced to a GFD, but without a clear association with compliance.

A major contribution of the present study is the representative sample, in which all children with T1D in our region were screened for CD at T1D diagnosis and followed up with up to ten years after T1D diagnosis, but several limitations also merit mentioning. First, there are no clear recommendations from the Food and Drug Administration (FDA) on how to best measure compliance, and previous studies have used different methods. Outcome measures could be based on histology, different serology, and clinical outcome assessment (including patient-reported outcomes) All these methods have advantages and shortcomings [21]. Newer measures, such as immunological tools [45] and gluten immunogenic peptides [46,47], have been reported to show a higher reliability. It would have been an advantage to also measure GIP in our study since it has been shown to be more reliable than tTGA in monitoring response to a GFD [46]. Unfortunately, this method was not yet established in clinical practice during the timespan of our study. tTGA is an indirect measure and has been shown to correlate well with mucosal damage of the intestine [19,20,48] and with the severity of the disease [49]. However, a recent meta-analysis found that tTGA often underestimates the degree of mucosal damage [50] and that normalized, complete histological recovery may not be obtained [51]. Another problem with tTGA is a partly individual response. Different patients digesting the same amount of gluten produce different amounts of tTGA [52], and at different speeds [53]. A small group of patients even became immune to gluten after longer GFD treatment and then did not produce tTGA at all [54]. Nevertheless, tTGA is recommended as a key outcome measure by the Tampere recommendations in 2018 [21]. Questionnaires assessing self-reported compliance in combination with tTGA are used by some and could have benefited our study. Another limitation is the absence of an age- and sex-matched group of children with CD only, which would have strengthened the comparison of compliance to a GFD in T1D children with CD compared to children with CD only. However, we have good estimates on compliance to a GFD from screened children with CD in Sweden [18,26], making our indirect comparisons valid.

## 5. Conclusions

We report a fairly high compliance to a GFD in Swedish children with T1D and CD, although compliance levels may be somewhat lower than in children with CD only. Moreover, we found older age and poorer metabolic control to be associated with poorer compliance. Our results indicate that children diagnosed with both T1D and CD could need intensified dietary support, especially the subgroups with older children and children with poorer metabolic control. 

## Figures and Tables

**Figure 1 nutrients-13-04444-f001:**
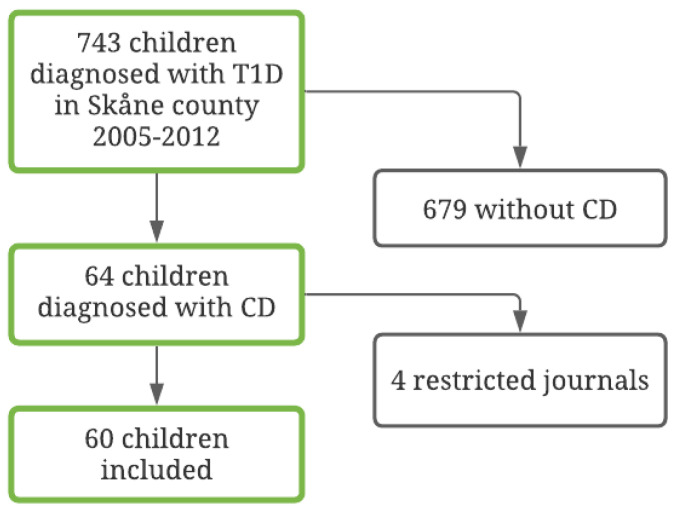
Flow chart of the included diabetes population, 2005–2012.

**Figure 2 nutrients-13-04444-f002:**
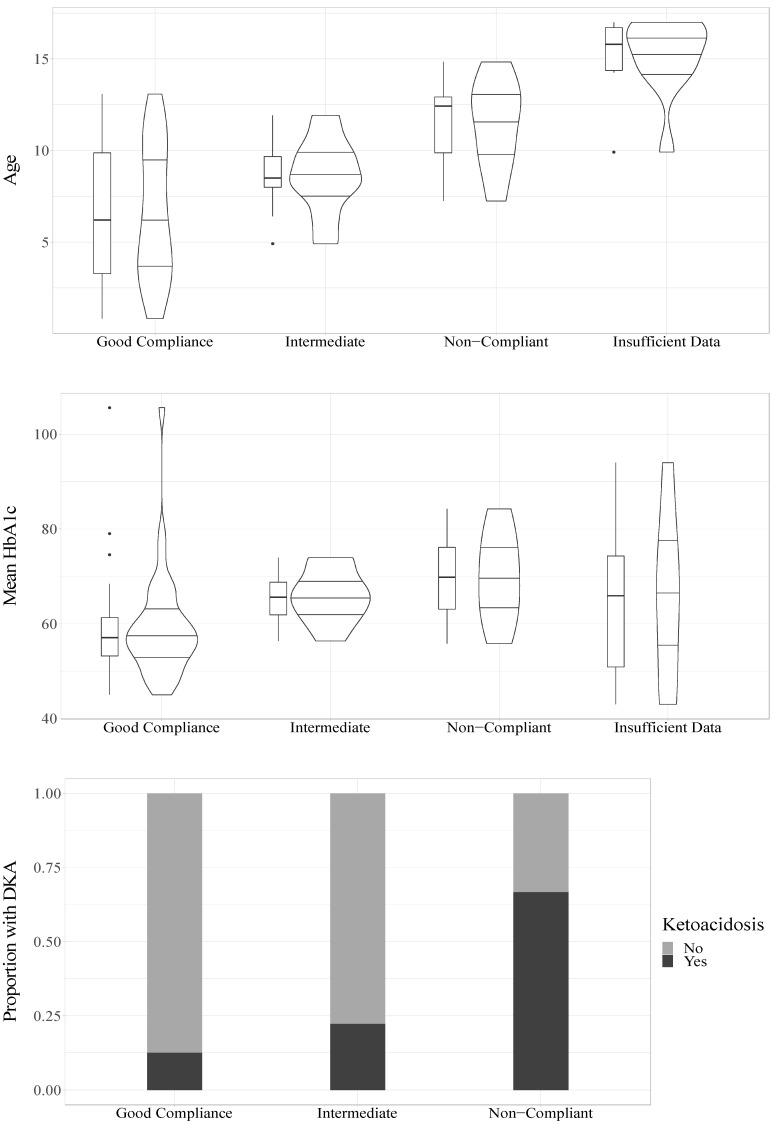
Relations of age, HbA1c and DKA to compliance.

**Table 1 nutrients-13-04444-t001:** Demographic and clinical variables in the full sample and across compliance subsamples.

		Subsamples Based on Compliance Classification			
	Full Sample (*n* = 60)	Good (*n* = 34)	Intermediate (*n* = 9)	Non-Compliant (*n* = 7)	Insufficient Data (*n* = 10)
Age at T1D diagnosis, M (SD)	8.87 (4.50)	6.54 (3.68)	8.63 (2.13)	11.44 (2.58)	15.18 (2.14)
Female, *n* (%)	32 (53%)	16 (47%)	6 (67%)	6 (86%)	4 (40%)
CD prior to T1D, *n* (%)	19 (32%)	12 (35%)	2 (22%)	3 (43%)	2 (20%)
Ketoacidosis, *n* (%)	10 (20%) ^a^	4 (13%) ^b^	2 (22%)	4 (67%) ^c^	0 (0%) ^d^
Mean HbA1c, M (SD)	62.36 (11.85)	59.37 (11.25)	65.95 (5.85)	69.77 (9.91)	65.31 (19.13)
Mean SDS-BMI, M (SD)	0.72 (1.14)	0.65 (0.78)	0.48 (0.95)	1.30 (2.50)	0.76 (0.76)

Notes. ^a^ Missing data for 9 participants. ^b^ Missing data for 2 participants. ^c^ Missing data for 1 participant. ^d^ Missing data for 6 participants.

## Data Availability

Data can be found in SWEDIABKIDS (https://ndr.nu, accessed last on 31 December 2020) and The National Patient Register—Socialstyrelsen (https://www.socialstyrelsen.se, accessed last on 31 December 2020).

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
