# Peer review of "Compliance to a Gluten-Free Diet in Swedish Children with Type 1 Diabetes and Celiac Disease"

_nutrients, 2021, doi:10.3390/nu13124444_

Round 1
Reviewer 1 Report
The paper is good, fluent and well-written. The data is retrospective so it has problems but data is also lacking on this subject. I think the authors should bring up the fact that T1D+CD compliance was lower (especially if you consider the intermediate compliance also poor) and that these patients need additional monitoring. The authros could also suggest what this additional monitoring could be. Below are also some minor comments:
-could the title be shortened, maybe remove "associations with age and metabolic control". Shorter is better.
-in introduction the authors say that tTGA is strongly related to the degree of intestinal mucosal. There is correlation but it is only moderate as serology normalises much faster than biopsy. I would remove the word strongly.
-I think the first sentence of discussion should be changed. For me the big finding is that compliance is poorer in T1D+CD in comparison to just CD. This should be brought up instantly, perhaps the big discussion can be later on.
- In the last chapter the author mention GIPs as a possibility. These have in my opinion shown good result, I believe they are rather well validated? Not much used but should be used more in clinics. I would remove the need for further validation and maybe just say that they have not entered clinics yet.
- In the conclusions, it is talked about that T1D+CD would need extra-attention, however, this is not talked in the actual discussion. Please discuss the need for extra-attention regarding GFD monitoring also in the discussion. We have also many new drugs in the pipeline and T1+CD could be one focus group of drug treatment in the future as the compliance is lower? Maybe not children first but still. See Schuppan et al. NEJM 2021 A "Randomized Trial of a Transglutaminase 2 Inhibitor for Celiac Disease".
- In the compliance chapter I would add that even if serology normalises the histological recovery may not be obtained. In the paper by Daveson et al. majority of the patients with well-controlled CD and serological recovery did not obtain mucosal recovery! (GastroHep 2019, "Baseline quantitative histology in therapeutics trials reveals villus atrophy in most patients with coeliac disease who appear well controlled on gluten-free diet").
- It would be interesting to do a prospective study on T1D+CD patients and study also their biopsies vs just CD controls, maybe next this? -These were all children so propably there were no control biopsies taken?
Author Response
Dear reviewer,
thank you for your positive response and most valuble and insightful comments.
We agree with your major comment about the importance of our finding that compliance in children with T1D+CD seem lower than in children with CD only and have highlighted this in our revision. We have also added suggestions on how treatment for these children could improve, possible including trials with transglutaminase 2 Inhibitors.
We also agree with most of the minor changes suggested and have therefor shortened the title, changed the word strongly concerning tTGA:s relation to mucosal damage, added some thoughts about GIP:s and removed the sentence about GIP:s needing validation. We have also added the references mentioned by Schuppan et al and by Daveson et al.
We agree that it would be most interesting to do a prospective study on T1D+CD patients vs CD controls and perhaps measure both tTGA and GIP in stool as well as biopsies. Maybe next time!
Again thank for your comments which we believe made our paper a lot better!
Best wishes,
Hanna Söderström
Corresponding author
Reviewer 2 Report
The study by Söderström is aimed to examine compliance to a GFD in children with both T1D and CD, by measuraments of tissue transglutaminase IgA-antibodies. Compliance to a GFD in CD patients is a very interesting and timely topic. However, the methodology used in this study is not appropriate to explore this issue. Indeed, it is largely recognized that serology (i.e., anti-tissue transglutaminase antibodies, anti-deamidated gliadin pepetides antibodies or anti-endomysial) shows poor sensitivity to identify all patients who consume gluten and fails to detect dietary transgressions in children and adolescents with CD; therefore serology is not a recommended tool to assess the adherence to a GFD (Comino I, Fernández-Bañares F, Esteve M, et al. Fecal Gluten Peptides Reveal Limitations of Serological Tests and Food Questionnaires for Monitoring Gluten-Free Diet in Celiac Disease Patients. Am J Gastroenterol 2016;111(10):1456-65). Moreover, the small sample size and the lack of a control group of patients with CD alone does not allow drawing conclusions on the compliance of patients with CD and T1D as compared with patients with CD alone.
Author Response
Dear Reviewer,
Thank you very much for reading our manuscript and give us your opinions.
We do agree that it is a limitation that we did not measure GIP in the stool since it has been shown that tTGA serology is less sensitive and therefore not as good to measure the compliance to a GFD. It would have been interesting if we would have data on GIP to compare the tTGA data with but unfortunately we do not. To our knowledge GIP is just beginning to enter into clinical practice and while this is a retrospective study GIP has not been measured. We had already included the reference you referred to but believe you were right to give it bigger attention and have tried to highlight this more in Limitations in the study.
Regarding our sample size we agree, but it is bigger than most studies we know with a well defined T1D population.
Concerning a control group we have mention this big limitation and that we refer to another of our studies, evaluating compliance with tTGA serology, a Swedish study with a screened group of children with CD, see below, which to our knowledge is a very comparable group of children since they also have a screening detected CD as do children with T1D. It would as you point out, of course have been better with a regular control group of children with screened CD only, but is very time consuming and extremely expensive to screen healthy children for CD.
Again thank you for helping us with our manuscript,
Sincerely,
Hanna Söderström
corresponding author
High adherence to a gluten-free diet in adolescents with screening-detected celiac disease.
Webb C, Myléus A, Norström F, Hammarroth S, Högberg L, Lagerqvist C, Rosén A, Sandström O, Stenhammar L, Ivarsson A, Carlsson A.J Pediatr Gastroenterol Nutr. 2015 Jan;60(1):54-9. doi: 10.1097/MPG.0000000000000571.PMID: 25238121
Reviewer 3 Report
Interesting topic.Good presentation
Author Response
Dear Reviewer,
Thank you very much for your positive response!
Best wishes,
Hanna Söderström
corresponding author
Round 2
Reviewer 2 Report
Dear Authors,
thank you for your reply and for including my suggestions in the revised manuscript. The manuscript is now suitable for publication.
Best regards